# Ultra-Orthodox Lesbian Women in Israel: Alternative Family Structures as a Bridge between Religious and Sexual Identities

**DOI:** 10.3390/ijerph19137575

**Published:** 2022-06-21

**Authors:** Shirley Ben Shlomo, Ayelet Oreg

**Affiliations:** The Louis and Gabi Weisfeld School of Social Work, Bar Ilan University, Ramat Gan 52900, Israel; ayelet.oreg@biu.ac.il

**Keywords:** family structures, identity, lesbian women, social representation theory, ultra-Orthodox community

## Abstract

The Jewish ultra-Orthodox community enforces strict rules concerning its members’ way of life and demands that their identities be consistent with that of this conservative community. However, such congruence does not exist for ultra-Orthodox women who identify as lesbians. Drawing on social representation theory, this study examines the unique family structures that lesbian ultra-Orthodox women in Israel have adopted to accommodate their conflicting identities. The study employed a qualitative multiple case study design, conducting in-depth interviews with seven ultra-Orthodox lesbian women, and adopted a phenomenological approach to learn about their lived experience. The women had all married young in arranged marriages and all had children. Four of them were still married, while the other three were divorced. In all cases, however, their lesbian identity was kept hidden. The findings reveal the unique family structures these women created that allowed them to maintain their religious way of life on the surface, while remaining committed to their sexual identity in secret. The study extends the social representation theory and promotes an understanding of the multifaceted identity of ultra-Orthodox lesbian women. The findings can aid in designing interventions that can help such women cope with the secret aspects of their life.

## 1. Introduction

One of the basic principles of collectivist societies is that the welfare and needs of the community take precedence over those of the individual [1] Thus, the collective identity, and the norms and behaviors it dictates, outweigh the personal identity. Consequently, any situation in which the welfare of the individual requires behaviors outside the accepted norms of the community is likely to generate harsh social stigma. Members of such societies are therefore often forced to suppress their personal needs in order to maintain their collective identity.

The Jewish ultra-Orthodox sector (also known as “Haredi”) in Israel is an example of this sort of collectivist society. As its core values are religion and family [2], calling either of them into question jeopardizes community belonging. The current study examines the lived family experience of lesbian women in ultra-Orthodox society who must navigate between their religious and sexual identities on the personal, couple, family, and community levels.

The study relies on social representation theory [3,4], which contends that identification with a group provides an individual with a sense of belonging and security in regard to the way they are meant to conduct themselves [5]. According to this theory, the individual’s social representations include their personal identity and identity with a group based on a system of shared beliefs, attitudes, and feelings. Social representations are constructed by means of a constant dialogue among the group members [6], and serve as a guide for action. At the heart of the system are values, which create and define the groups’ shared goals [3]. Holding multiple social representations is known as *cognitive polyphasia* [7], and is especially characteristic of modern society, which acknowledges that an individual can maintain several identities that do not necessarily coincide with one another. In contrast, in conservative societies, multiple identities are only possible when they are internally consistent. Conflicting identities create an obstacle to relations between the individual and the group. In order to continue to belong to the group, they must choose which of the identities takes precedence. However, choosing one identity at the expense of other entails erasing significant aspects of the individual’s self-identity, which may, in turn, lead to emotional distress [8]. The current study focuses on the alternative family structures created by lesbian ultra-Orthodox women in Israel to bridge the gap between their sexual and religious identities in order to maintain their membership in the community.

### 1.1. The Ultra-Orthodox Community in Israel

The ultra-Orthodox sector is a distinct Jewish community characterized by fundamentalist religious beliefs [9]. Most of its members reside in Israel, constituting 11% of the population [10], with large ultra-Orthodox communities in the U.S. and Europe as well [11]. Although to people outside this society, it appears to be a homogeneous and unified community, it is, in fact, composed of numerous groups who distinguish themselves from one another on the basis of religious attitudes and practices or ethnic backgrounds [12]. All these groups, however, share a conservatism aimed at preventing the incursion of modern secular society [13,14]. Unlike other religious communities in Israel and elsewhere, the ultra-Orthodox are committed primarily to religious law, spurning Israeli society in general, including the state and its laws [15]. The members of this sector regard themselves as subject not to the authority of the government, but to that of its rabbis, who dictate strict behavioral codes based on Halacha (religious law) in respect to every aspect of the life of the individual, the family [16], and the community [17].

The ultra-Orthodox isolate themselves from secular society, living in separate neighborhoods and operating their own system of education independent of the national school system. Ultra-Orthodox schools in Israel concentrate solely on religious studies in order to preserve their insular society and protect it from external modern influences. Subjects of general knowledge are not taught [18], and there is a strict separation between boys and girls throughout the school years.

Members of this sector maintain rigid conservative dress codes and devoutly follow Halacha by means of rigorous religious practices [13,19], which serve as an external mechanism for the defense of the community’s boundaries. At the same time, internal control mechanisms are in place to control all aspects of the members’ lives, and particularly those pertaining to the community’s key values: religion and family [2]. Indeed, family and children are central to this society, with members marrying young in arranged marriages and aspiring to raise large families [20]. Marriage is regarded first and foremost as a business arrangement between the parents of the intended couple, the future spouse being chosen on the basis of criteria such as the specific religious group to which they belong and the family’s economic status, health, and piety, with no attention paid to the youngsters’ feelings [21]. Consequently, meetings with potential mates are focused on the objective of marriage, and dates for pleasure are considered futile and inappropriate [22].

### 1.2. Control Mechanisms on the Family

While maintaining the integrity of the family unit is the responsibility of the couple, the extended family and community are also involved in ensuring that this central value of family life is upheld. Whereas it is the parents who are tasked with arranging the marriage, they are joined in controlling the couple’s life after the wedding by specially designed rabbinical and counseling systems. Thus, for example, as divorce represents a threat to the family unit [23], if rifts appear between the couple, both the parents and community leaders make every effort to prevent the dissolution of the marriage. Divorce is considered a personal, family, and social failure. It brings with it a social alienation from neighbors and official community authorities, children of divorced parents often being banned from their schools, and overall, divorce within the family, severely harming the chances of arranging a “good” marriage for the children in the future. These considerations typically carry more weight than the potential implications of conflicts between the couple on the mental health of the parents and children. As a result, even in severe cases, such as spousal violence, the integrity of the family is likely to take precedence over personal welfare [24]. The same is true when there are signs of sexual abuse within the family. Even in this situation, ultra-Orthodox society generally prefers to preserve the unity of the family rather than exposing the molester [25]. These examples demonstrate a primary feature of collectivist societies such as the ultra-Orthodox, in which the welfare and needs of the community take precedence over those of the individual [1]. On the whole, ultra-Orthodox society is characterized by a “culture of silence”, particularly in regard to sexual issues [26]. Should an individual experience distress in the family, or desire to reveal a problem such as abuse, they are expected to employ religious strategies, namely, prayer and faith in God [25], rather than strategies of exposure or disclosure. The same rule applies in the case of gay or lesbian identity. Such a sexual orientation can be defined as a stigmatized identity, that is, an identity that carries a social stigma and therefore cannot be revealed. The price of this lack of disclosure is considerable mental distress [27].

In order to prevent the intrusion of others into sensitive matters, such as sexual issues, ultra-Orthodox individuals or couples turn to their rabbi for advice, and, if he deems it necessary, he refers them to a counselor in the community who deals with the particular problem [28]. Sexual relations before marriage are strictly forbidden, and, as noted above, boys and girls attend separate schools to eliminate the possibility of any interactions between them. In addition, behavioral rules of modesty are firmly upheld, including avoidance of any public reference to sexuality, and youngsters are paired at a young age through arranged marriages rather than being given the opportunity to choose their spouse [29].

Furthermore, the control mechanisms in ultra-Orthodox society perpetuate the lower status of women in comparison to men. Women exist on an intermediate plane somewhere between the public and the private spheres. In most cases, it is the woman who provides for the family, while her husband devotes himself entirely to religious studies [30] in a group setting, in line with the attitude that “religious study is his trade.” At the same time, women are also expected to play a dominant role in the private sphere, performing all the household duties and raising the children. Thus, women work, their husbands study Torah, and the cost of his studies is shared by the couple. This way of life requires women to dedicate themselves entirely to the spiritual advancement of men. Consequently, a young ultra-Orthodox woman learns that the most significant goal in her life is to be the wife of a religious scholar, whose studies are his life and whose sons will go on to be religious scholars like him.

Socialization for this role begins at an early age when girls watch their mothers and help in the housework and in caring for their younger siblings. It continues in their all-girls schools, which serve as external agents of socialization for family, emphasizing the pupils’ role in the private sphere and encouraging it by extended vacations during religious holidays and not assigning homework on the eve of the Sabbath and holidays [31]. Whereas boys are brought up to honor the value of religious studies, girls are brought up primarily to honor the values of family and sexual modesty [18].

### 1.3. Sex, Sexuality, and Sexual Identity

Girls’ encounter with sexuality generally begins when they get their first period, although menstruation is not associated with sexuality at this age. Interesting findings in this regard emerged from interviews with women from one of the strictest ultra-Orthodox communities in Israel. The women reported that their main sources of information about the menstrual cycle were their mothers and peers. While some described receiving only technical information, others said they were given positive messages concerning their future ability to bring children into the world. Nevertheless, all the interviewees stated that the subject of menstruation was taboo, meant to be kept private and not associated in any way with sexuality [32].

Ultra-Orthodox couples regard sexual relations as an act of holiness, just like every other human activity, and therefore they are to be accompanied by thoughts of their holy nature [15]. The society’s norms also include instructions as to the appropriate times for this activity. The most significant is seven days following the start of the wife’s period, when she is required to purify herself in the mikve (ritual bath) and resume intimate relations the same night after the halt imposed during menses. Halacha expressly forbids forcible sexual behavior of any sort. However, refraining from sexual relations on “mikve night” is allowed only in exceptional circumstances. Lack of willingness or desire on the part of either member of the couple is considered non-normative behavior reflecting personal or couple problems that demand professional intervention. Another significant time for sexual relations is on the eve of the Sabbath, when sexuality and holiness are seen to come together [33].

Against this background, it is clear that lesbian or gay sexual identity has no place in ultra-Orthodox society. Although no data is available regarding the number of LGBT individuals in this sector, in a representative sample of the Jewish population aged 18–44 in Israel, it was found that 11% of the men and 15% of the women reported being attracted to members of their own sex. The authors contend that it can be assumed that the distribution of in the ultra-Orthodox community is similar to that in the general population, the only difference lying in the manner in which less accepted sexual identities are coped with in this society [34]. 

The experience of lesbian ultra-Orthodox women navigating between their religious and sexual identities, the focus of the current study, has never previously been systematically investigated. However, more and more articles on the subject have recently been appearing in the popular media. Another relevant source of information is the Bat-Kol organization, founded in 2005, which provides support for women who, according to its mission statement, “are not willing to give up their religious identity or their right to express their identity as lesbians.” Most of the academic literature on LGBT individuals in the Jewish community comes from the U.S., where 7% report defining themselves as LGBT. Margolis [35] claims that these people face potential discrimination from several sources, including family, society, and religion. One of the major difficulties they must contend with is the fact that they are a double minority, that is, a minority in terms of both their ethnicity-religion and their sexual orientation [36], a status that may lead to stress, confusion, and anxiety [37]. It is important to note that while the process of solidifying one’s sexual identity is a major step toward achieving mental health, the individual’s family and cultural context do not always enable its disclosure [38]. Being gay or lesbian in ultra-Orthodox society is an example of such a situation. A study conducted among gay men in this community may also shed light on the impact of the religious, cultural, and social context of lesbian ultra-Orthodox women [39]. The participants noted that as the ultra-Orthodox community was the only one they knew, they could not imagine a life outside it. In addition, they reported choosing to keep their sexual identity a secret in order to avoid the negative consequences they and their family would suffer should it become known. Interestingly, it was found that although the participants bore two conflicting identities, the religious identity was the core of their self-definition and therefore took precedence over the sexual identity. As a result, they attempted to change their sexual identity, and when their efforts were unsuccessful, they kept it hidden. Moreover, they did not view their nuclear family as a source of support, believing that they would turn their back on them if they discovered their secret so as not to jeopardize the possibility of finding a good match for their siblings. Consequently, the men felt very much alone [39]. A small number of American studies have also investigated the experience of lesbian women in ultra-Orthodox communities and found a similar conflict between their desire to express their sexual identity while still preserving their religious identity [36,40].

The current study is the first to consider lesbian ultra-Orthodox women in Israel in an effort to learn about the personal, couple, family, and community experience of these women. Furthermore, it focuses on an aspect of their experience that has never previously been investigated, namely, the manner in which they bridge the gap between the conflicting social representations of religious and sexual identities, with an emphasis on family life and the unique family models they have constructed.

## 2. Method

### Procedure and Data Analysis

The study was conducted during 2021, after being approved by the Ethics Committee of the School of Social Work at Bar-Ilan University. It was conducted as qualitative research with an interpretative paradigm. Thus, the positionality of the researchers is of significance, since the subject under investigation is experienced and interpreted through their own world of content and identity. We identify ourselves as researchers who are women, mothers, social workers, Israeli, Jewish, secular, and heterosexual.

The study was guided by a central question: What is the personal, couple, family, and community experience of ultra-Orthodox lesbian women in Israel? In view of the many restrictions and strict boundaries in the ultra-Orthodox community, we adopted a qualitative, phenomenological multiple case study approach [41]. This methodology facilitates empirical exploration of people’s shared experiences of a unique phenomenon, in our case the conflicting identities of being ultra-Orthodox (and preferably a “good” ultra-Orthodox woman) and lesbian, and enables close examination, both within and across cases. While all the women in our study shared the same phenomenon and research context, they reported unique individualized experiences [41]. The study also displays the features of a revelatory case study, as it is the first to explore the phenomenon of negotiating conflicting female identities within the very rigid family structure in Israeli ultra-Orthodox society. 

In-depth interviews were conducted with seven women (each woman is a case study). All women in our multiple case study design define themselves as ultra-Orthodox, Lesbians and observing religious laws and restrictions. 

The participants were recruited through the Bat-Kol organization—a religious lesbian organization, founded in 2005, by a group of women who were not willing to relinquish their religious identity or their right to live as lesbians. The organization supports ultra-Orthodox lesbians, particularly those who have not come out, by offering social activities and a supportive environment. Bat-Kol’s mission is also to provide education geared toward the integration of lesbians in the religious community (for more information about Bat-Kol, see Available online: http://www.bat-kol.org/english/ (accessed on 12 May 2022)). Due to the sensitive nature of this study, recruiting participants was very challenging. We had a prior acquaintance with an active member at Bat-Kol and she had introduced us to her friends. We used “snowball sampling” for recruiting the seven women whom we ended up interviewing. Two of the women had initially asked that we use pseudonyms. Following the interview, once we gained these women’s trust, they had agreed to share their real names. 

A multiple case study have from 4 to 10 cases, each of which should be situated within the unique contexts that provide the information to be analyzed [41]. Accordingly, our study contains seven cases. Informed consent was obtained from all women involved in the study. In order to develop a deeper understanding of their unique experiences, we conducted both within-case and cross-case analysis. Cross-case analysis involves multiple levels ([41]. First, we analyzed the individual content of each interview. In the next stage, we looked for the commonalities and differences between the seven cases to facilitate an inclusive analysis. In addition, to gain a deeper understanding of the women’s lived experiences, we conducted content analysis using the phenomenology approach. Throughout the analysis, we focused on the women’s own descriptions of their experiences [42]. The characteristics of the sample appear in Table 1.

## 3. Results

Cross-case analysis revealed five main themes.

**Theme 1: The Social Construction of the Religious Identity.** This theme deals with the mechanisms creating the social construct of ultra-Orthodox religious identity and can be divided into three sub-themes: the nuclear family, the educational system, and arranged marriages. All these serve as agents constructing the individual’s social representation and are described as significant to the woman’s life and the socialization she underwent that shaped her personal identity in general and her sexual identity in particular. In effect, there are two pathways to the construction of identity in ultra-Orthodox society. The first leads to a situation in which the woman’s personal identity conforms to her family and community identity, resulting in consistent identities. This situation is irrelevant to the women in our sample who followed the second pathway, in which the woman’s personal–sexual identity is inconsistent with her family–community identity, engendering severe internal conflict.

**Theme 2: The Family and Educational System: Ignorance, Secrecy, and Repression of Women’s Sexuality.** This theme relates to the development of the woman’s sexual identity, and demonstrates how discourse, or lack of it, on sex and sexuality affect the woman’s perception of this issue and the way in which her sexual identity is solidified. Two sub-themes emerged here: sexuality in the family of origin, and sexuality in the all-girl’s high school (known as a seminar). In both cases, the participants referred to what is apparent on the surface versus what goes on under the surface. 

**Theme 3: The Couple: Overt and Covert.** This theme deals with the revealed and concealed aspects of coping with the woman’s conflicting identities within the couple relationship. In terms of master narratives, the women described a clash between the narratives they tell themselves about themselves, one concerning their desire to be an ultra-Orthodox wife and mother according to the norms in their society, and the other regarding their lesbian identity and desire for a relationship with a female partner. The discord between narratives was expressed in two sub-themes: the wedding night and intimate relations with their husband, and their internal conflict.

**Theme 4: Children, the Extended Family, and the Community: Disclosure and Concealment.** This theme describes aspects of the complexity of the woman’s daily life in the context of family and community, whereby she maintains the outward appearance of an ultra-Orthodox wife and mother while keeping her personal and familial conduct as a lesbian secret. This complexity was manifested in three sub-themes: keeping secrets from the children, parents, and siblings; keeping secrets from the community or asking for help from leading community members; and contact with the Bat-Kol organization.

**Theme 5: Alternative Family Structures: Resolving the Conflict.** Two sub-themes described different family models: the woman remains married but has two couple relationships (Cases 2, 3, 4, 6); or the woman divorces but keeps her lesbian relationship secret (Cases 1, 5, 7). These structures offer a solution by enabling the woman to live with her two conflicting identities within the context of the complexities of couple, family, and community life in ultra-Orthodox society. 

### 3.1. Theme 1. The Social Construction of the Religious Identity

Construction of the ultra-Orthodox identity begins at a very early age and runs throughout the social constructs, from the family to the schools and the community.

#### 3.1.1. Family of Origin

The family of origin significantly influences solidification of the individual’s value system and identity in general, and in conservative ultra-Orthodox society in particular. The interviewees described a variety of experiences that shaped their childhood, adolescence, and who they are today. They came from a diverse range of families: native-born Israelis who returned to religion (The term “return to religion” (hazara be-tshuva) is used for a secular Jew who chooses to embrace religious faith and lead a devout life, observing all the religious commandments and practices); families from Europe and the U.S. who returned to religion; and families belonging to a variety of Hasidic sects. The inherent hierarchy in ultra-Orthodox society between families of greater or lesser distinction also played a role in the women’s perception of their religious identity and community belonging.

The desire to be part of a community and adopt its values in childhood and adolescence ran like a thread through the interviews. For many of the woman, cracks had already opened in their sense of belonging when they were younger, for example, if they grew up in a family that had returned to religion or in a family that has Sephardic origins (e.g., versus Ashkenazi origins), which is considered lower in the hierarchy within the ultra-Orthodox society. Moreover, their comments revealed that when their sexual identity began to take shape, the sense of belonging to the community that had developed in early childhood was undermined and they only seemingly belonged, although their inner faith and connection to the religious world was not affected. This can be seen in the following remarks:

“*I was ashamed of my parents…As people who had returned to religion they didn’t understand how hard it was for me outside. People who grow up in ultra-Orthodox society understand nuances of the society that my parents were unaware of*”.(A., Woman no. 1)

“*My parents welcomed all the lepers and good-for-nothings into our home. They didn’t agree to regard them as second-class the way the rest of society did. My mother was a lawyer, a profession she acquired when she was secular, and she used to represent Sephardic Jews who weren’t accepted into Ashkenazi institutions. My parents were proud of bringing us up to see everyone as equal*”.(M. Woman no. 5)

“*Belonging? I belong to ultra-Orthodox society, totally. I’m more pious than I used to be, but I’m different from the average ultra-Orthodox*”.(B., Woman no. 2)

She went on to explain that her difference lay in her attraction to women.

#### 3.1.2. Educational System

The strict system of religious laws and practices includes clear messages regarding “right and wrong” in respect to sex and sexuality. According to all the interviewees, the schools play a central role in conveying these messages and in shaping their religious and sexual identities.

“*The educational world I grew up in is characterized by very dichotomous values, reward and punishment, good and evil, and there was a lot of concern for modesty: the length of the sleeve, the length of the skirt, concealment*”.(B., Woman no. 6)

“*There was no sex education. They don’t talk about it in school because it doesn’t exist.The primary value in the schools I went to was to preserve modesty, to dress modestly, oh, and not to use bad words. To this day I don’t know how to curse*”.(A. Woman no. 1)

#### 3.1.3. Arranged Marriages

The interviews reveal that the climax of the process of socialization for couplehood and family, as well as adoption of the ultra-Orthodox identity, comes when it is time to find a mate. Matchmaking is on the order of a symbolic community seal of approval for the religious status of the young man and woman, as well as their families. Parameters relating to the family of origin, such as if they returned to religion or were born in the community, if there is a disability or genetic defect in the family, if the family is of Sephardic or Ashkenazi origins, if there are renowned religious scholars or rabbis in the family, are all taken into account in the choice of a suitable mate. Considerable weight is given to the perception of the normativity of the family and their observance of all the religious laws and practices. In addition, the personal qualities of the young person are scrutinized: Is he a good religious scholar? Is she modest enough? No thought is given to emotional or romantic considerations.

“*Among the ultra-Orthodox it is not legitimate to talk about love because you don’t marry for love. People marry because they are considered a good match and have the same approach to raising children, whether it works in the end is a matter of luck*”.(M. (Woman no. 4)

The story of how the women were matched with their husbands occupied a central place in all the interviews. This was a significant point in their life, the point at which they took on themselves (or in some cases were forced to take on themselves) the identity of a religious woman, which dictates their outward conduct, that is, the “dos and don’ts” expected from a wife in ultra-Orthodox society. All the women in the sample had had relations with other girls in their seminar before their wedding night, whether sexual relations, sexual experimentation with friends, or romantic relations. For all of them, the matchmaking process was a sign for them to return to the right social/community path, and they all described being compelled to part from their female partners or intimate friends as a crisis in their lives.

“*I got married when I was nineteen and a half. It was an arranged marriage. I met with a few guys and what was important to me was a good family and a nice man. When I saw my husband, he seemed okay. Not good, not bad*”.(A., Woman no. 1)

“*I lived a parallel life—meetings with potential mates and relations with girls. It was like a game. I enjoyed it that men liked me*”.(M., Woman no.5)

The women also had to contend with conflicts surrounding future matches, that is, when their own children reached marriageable age. All of them referred to the fear that they would impair their children’s chances of making a good match, and it was this fear that dictated their current family structure. On the surface, they lived a normative, conventional ultra-Orthodox family life while in secret they were also in another relationship, thereby meeting the different sexual and emotional needs of themselves and their husbands. The issue of their children’s future matches was referred to repeatedly.

“*If I make a move to leave [the marriage] and say outside the borders of my family that I like women, I won’t remain in my society and that will hurt their matchmaking chances*”.(B., Woman no. 2)

“*I created a whole world for myself that isn’t the society I came from and that’s wonderful, but I always go back there so as not to hurt the children’s future matches and the schools they go to*”.(H., Woman no. 3)

### 3.2. Theme 2. The Family and Educational System: Ignorance, Secrecy, and Repression of Women’s Sexuality

As noted above, modesty is a major value in the ultra-Orthodox woman’s socialization. From an early age, girls are given messages regarding the importance of hiding their body and obscuring any sign of their femininity. At the same time, there is a total absence of any reference to sexuality. Although this is beyond the scope of the present research, it is worth noting that in recent years media and rabbinical discourse in ultra-Orthodox society has become more radicalized, restricting the public space of women more than ever. The messages the girls receive are conveyed by the schools, synagogues, and *pashkevils* (Pashkevils are broadsides or posters pasted to public walls in ultra-Orthodox communities to provide the residents with information, often regarding proper conduct. They are a major means of mass communication in these communities), in which immodesty is often said to blame for epidemics, calamities, and disasters.

#### 3.2.1. Lack of Discourse on Sexuality

All the interviewees stated that they had witnessed no open discussion whatsoever of sexual development, sexual relations, sexuality, or the opposite sex during childhood or adolescence. The subject was raised for the first time, albeit in a limited fashion, by the woman in the community who is responsible for preparing brides for their wedding night.

“*Sex education in the society I grew up in—it’s nonexistent. There is only education for no sex: modesty, hiding that you’re developing, God will not love you if you’re not modest*”.(R. (Woman no. 7)

“*There was no mention of sex in my home or in school. The society is very conservative. When I was young I read an anatomy book that was in the house, probably from my parents’ life before they became religious. For the first time I was exposed to the subject of how babies are born. I went to my mother and asked her if I had understood it correctly and she said, “Righteous people don’t do that.” To this day I haven’t forgiven her for saying that*”.(M., Woman no. 4)

“*My parents have a neighbor who got engaged without knowing anything. You have to understand, that’s very rare because usually you get a vague idea from girlfriends. The woman who prepares brides tried to explain to her that there’s another issue of Halacha they had to talk about, but she wouldn’t listen. The woman spoke to the girl’s mother and no matter how appalling it sounds, they sat the father down to explain to the girl what she had to do with the man in terms of religious law. She cried for three days. It’s usually a trauma of one night. Today she has six kids. It’s bad for the boys too. They have to rape the bride when they really want to honor her*”.(M., Woman no. 5)

The vacuum created by the lack of discourse alongside the natural curiosity aroused by puberty is joined by the absence of familiarity with the opposite sex. As R. (Woman no. 7) related:

“*When I was a young girl I wasn’t allowed to go to homes where there were big brothers, boys. I didn’t understand what the problem was. Nobody talked about those things.*”

#### 3.2.2. All-Girls’ Seminar

In the seminar, the girls pieced together the little they had learned from older sisters or married friends regarding puberty, sexuality, and sexual relations. Nevertheless, their knowledge of these subjects was very limited. The interviewees described the narratives the girls created to explain the desires and fantasies awoken in adolescence and answer their questions about sex and their changing bodies.

“*I had an ultra-Orthodox girlfriend who told me this is what girls do because boys and girls are kept separated. I was actually sure that the term good friend meant a girlfriend you slept with*”.(M., Woman no. 4)

“*Before I got married I was in a relationship with a girl from the seminar. It came from lack of understanding, something pure. Those days there weren’t any Gay Pride parades. We didn’t understand what was happening…My family knew about my relationship with her. The seminar informed them. My family didn’t consider it worthy of their attention. It happens sometimes because of the girls’ naivete*”.(B., Woman no. 2)

The narrative describing a lesbian relationship as the product of “the girls’ naivete” appeared in other interviews as well. M. (Woman no. 4), for example, stated that when she told her husband about her romantic relationship with a girl before they were married, he responded, “Seminar girls and their nonsense.”

Despite the absence of any discourse or even familiarity with the term “lesbian” in ultra-Orthodox society, many of the interviewees reported feeling the need to hide the romantic relationships they had had in the seminar before marriage.

“*I didn’t even understand that I was a lesbian. I didn’t know the word. It didn’t exist in our world. But I guess I understood that we were doing something wrong because I didn’t tell anyone*”.(A., Woman no. 1)

“*When I was in high school I met a girl who was my first partner…We were 14, maybe 15. It developed into a sexual relationship very quickly, to the extent that there can be a sexual relationship between two ultra-Orthodox girls. I didn’t understand that it was a physical attraction and I had a lot of guilt. I kept telling myself I had to overcome it. My girlfriend called our relationship our “secret drawer.” Since we were also good friends, I told myself that any close friendship can become a relationship with kisses and hugs. It was a very immature bond, but one we couldn’t ignore. There was a lot of childish touching, without the understanding that it had anything to do with sexuality. Nevertheless, I understood that I had to hide it*”.(R., Woman no. 7)

The idea of a “secret drawer” was also expressed by another participant, who described the narrative she created in which her lesbian orientation was like an imaginary friend. In theoretical terms, it can be said that her initial fantasies about boys (derived from internalization of the strict social codes) were replaced by fantasies about girls, a safer place for childhood fantasies and dreams. As she grew older, this world, too, was shaken when she began to realize that her behavior was not in line with her society’s expectations from women.

“*I fell in love for the first time with a girl in ninth grade. I understood that she was more than a good friend, but I didn’t understand anything beyond that. I think over the years my lesbianism became like an imaginary friend. As a kid I had an imaginary friend for many years and I’m quite sure I replaced him with fantasies about girls. Before I fell in love at the age of 14, I had a fantasy about a boy. It scared me a lot because I know it was a thought I had to get out of my mind, and then when I fell in love with a girl in my class I felt it was permissible. At least they didn’t say it was forbidden. It’s regarded as a friendship*”.(M., Woman no. 4)

### 3.3. Theme 3. The Couple: Overt and Covert

The need to hide certain aspects of their identity while revealing others continued to be part of the participants’ lives, reaching its height once they were married. Stories of the wedding night and intimacy with their husband were found in all the interviews. Each of the women reported difficulty with sexual relations and feelings of aversion, longing for their secret friends or partners from the seminar, guilt, confusion, and sadness.

#### 3.3.1. The Wedding Night and Intimacy with Their Husband

According to most of the women, their first sexual encounter with their husband on their wedding night aroused difficulty and aversion to the sex act. It appears that the physical difficulty was given overt expression while the emotional difficulty remained a secret that became harder and harder to bear in the course of the marriage.

“*The first night after the wedding, which is the climax of couplehood, we didn’t have sex…That night I fell asleep and dreamed I was doing it with her. She was at my wedding, too*”.(B., Woman no. 2)

“*The first night was okay, but later every time he touched me I would recoil. It went on like that for many years; touch, foreplay were terrible experiences for me. I did everything for it to work and then one night when I was with my husband I called out the name of the friend I had had while we were having sex. My husband didn’t react, as if he hadn’t heard. I got very scared, and that very day I cut her presence out of me … At a certain point I developed vestibulitis, and that killed our sexual relations and our relations in general*”.(M., Woman no. 4)

#### 3.3.2. Internal Conflict

The women described undergoing a long and painful inner process after they were married in order to resolve the conflict between their lesbian and ultra-Orthodox identities. At this point in their lives, their religious identity was not only their personal identity, but also part of the fabric of their couple and family life. The interviewees spoke of their efforts to repress their sexual identity and the pain and distress they experienced in this period.

“*There were years of terrible loneliness in my marriage, even though we were friends. I wondered why it was so hard for me if I was a married woman and did what God wanted from me. During the prayers on Yom Kippur I would apologize to God for my orientation, I’d talk to God about how tired I was of feeling the way I did, I’d ask Him why I was forbidden to pray for a female partner. Sometimes I’d get angry and say “If You can’t accept my true prayers I don’t want to pray to You.” Actually, I was debating with myself*”.(G., Woman no. 6)

A subtheme of this motif related to the budding of the alternative family structures addressed in Theme 5. At this point, however, the women still did not have a solidified lesbian identity and were keeping their sexual orientation a secret from their husbands, and in certain cases from themselves as well. Consequently, they were struggling internally with their attraction to women or attempting to repress it.

“*My former partner got married in an arranged match like me. She lived near me, and at a certain point we started to invite her and her husband to Friday night dinner. For the first few years, even when I saw her at dinner I didn’t think about our previous relationship. I was preoccupied by my home and myself. My first partner and I are in contact. Not physical, just friends, and once when we got together I told her I thought I was asexual. She laughed. I didn’t understand why. I had erased everything in the past*”.(R., Woman no. 7)

“*My relationship with my current partner went on for several months after I got married, but it was she who asked to stop. It came from a religious place. Everyone gets their turn, and it wasn’t her turn yet*”.(B., Woman no. 2)

### 3.4. Theme 4. Children, the Extended Family, and the Community: Disclosure and Concealment

The emotional distress described in the previous theme took a daily toll not only on the couple, but also on other family members. M. (Woman no. 4) summed this situation up concisely: “That’s how it is with us. You get used to living with secrets.”

The secrets serve to protect the outward appearance of a normative ultra-Orthodox family, preventing the risk of discrimination against the children in the schools and damage to their future matchmaking. Diverging from the norm (divorce, coming out, or even using a cellphone) are a cause for the community to disqualify the family. In order to protect the family and themselves, some couples chose deliberately not to share the wife’s sexual orientation and the arrangements they had made to accommodate it with their children, siblings, parents, or other members of the community.

#### 3.4.1. Keeping the Secret from Children, Parents, and Siblings

The participants and their husbands made considered decisions as to who and what to tell the people around them. Some chose to convey an edited version of the situation to their children.

“*My 14-year-old daughter knows about me. She accepted it cheerfully and with love. We explained it to her, my husband and I together, when she was eleven. We didn’t talk about sex but just said that I love women in addition to loving her father*”.(B., Woman no. 2)

In other cases, the children were unaware of their mother’s secret.

“*It would hurt my children the most. They know I have girlfriends, but I made the conscious decision not to tell them [about her sexual orientation] because they go to an ultra-Orthodox yeshiva. If there were rumors, I’d deny them, and they’d do the same, and at least they wouldn’t feel they were lying. That’s how it is with us, you get used to living with secrets. We don’t talk about the divorce either, but we can’t hide that because we live in separate homes*”.(M., Woman no. 4)

Sometimes the secret is also kept from the couple’s siblings and parents.

“*My brothers know, my parents don’t. I mean I never told them and I won’t. But I’m sure my mother knows, I have the feeling she does. But we don’t talk about it*”.(A., Woman no. 1)

The remarks of one woman’s brother may be an indication of the narrative the extended family, and perhaps the community at large, has devised in respect to lesbian women.

“*My brother is a serious religious scholar. One day we were sitting down together and he said, “You know, B., we have the same taste in women.” The family doesn’t regard me as a lesbian, but as a woman who likes women*”.(B., Woman no. 2)

All the interviews contained reference to the emotional difficulty of living a secret life, reporting feelings of loneliness, sorrow, depression, fear of discovery, and despair.

“*I see my husband suffering so much and on the other hand he lives a life of deep faith and he has to keep it a secret. He has no one to talk to. No one would accept him. We’re both victims. Last week he got very angry and accused me and we’re sort of estranged now. I told him, I know it isn’t easy, but you’re telling yourself the wrong things. You can be depressed or you can say, it is what it is. You can also get up and walk out. You can do whatever is possible. He’s also a victim of a society that doesn’t accept and can’t contain. If he says anything we’ll be ostracized and he doesn’t have the courage to do something like that*”.(H., Woman no. 3)

#### 3.4.2. Keeping the Secret from the Community or Asking for Help

As in the case of the family, some of the women kept their secret from community members while others disclosed it, at least in part, to select individuals. The following examples illustrate the different strategies in light of the norms of conservative ultra-Orthodox society.

“*In the children’s schools, they don’t know. The truth is that there are no rabbinical instructions regarding relationships between women. In the past, when the issue was raised, the rabbis said there was no such thing. If it doesn’t exist, that’s good for us, but at the same time it means there are a lot of women who are trapped in their bodies and can’t get out. In the ultra-Orthodox community, if a kid comes to school with a phone that’s not kosher, in certain schools he’ll be kicked out on the spot. They wouldn’t know what to do about a lesbian relationship*”.(M., Woman no. 5)

“*Today I’m half in the closet. I started to tell secular friends. At my previous job no one knew. There are rumors in ultra-Orthodox circles. I don’t confirm and I don’t deny. In general, in our society there’s no problem with a woman who’s married and has a female partner. They regard her as a good friend, because there’s no such thing as a lesbian. They say you have a good friend and sometimes you hug her. Among the ultra-Orthodox it isn’t actually legitimate to talk about love because you don’t marry for love … There’s no legitimacy for a story like mine*”.(M., Woman no. 4)

The distress caused by the conflicting identities and the complexity of maintaining an alternative family structure (see below) led some of the couples to seek help and emotional support. However, the women reported harsh experiences with figures who are meant to be supportive but turned out to be useless at best and abusive at worst. This issue goes beyond the scope of the present study, but warrants attention in future research. As described by one participant:

*My husband and I decided to go to a counselor. I asked in the group and a friend of mine suggested a pious ultra-Orthodox woman who is a counselor and a teacher. I picked up the phone and I went to see her. At a certain point, my husband entered the picture. It was counseling that turned into terrible abuse. From her point of view, what I did was forbidden and I had to end the relationship [with her partner]. The situation got very bad. She would track what I was doing and call my husband. She told him to install cameras. It was a counseling of control. She cut me off completely from the woman. At that time, there was nothing happening with my husband either. It was a mess. She was the first one I told that I couldn’t sleep with him and that it was hard for me to pray because of what I did. Her counseling was to put surveillance on me so I’d change. Today I’m seeing a different counselor but it still hurts. It’s hard for me to talk about it. It reached very low. She forced my husband not to give in, to molest me, and that’s not in his character. In the two years I was seeing her I deteriorated into a very bad state. When she saw she was losing me she disappeared. Maybe it frightened her, I don’t know. She controlled me so tightly that I couldn’t manage without her. Right away I started going to a different counselor*.(H., Woman no. 2)

Some couples sought help from religious authorities, such as rabbis, also to no avail.

“*I looked for rabbis who would give me their approval [to be a lesbian]. Not one agreed to recognize love between women. There was just one who said, “If you do it you’ll be alone. They’ll take your children from you and you won’t have a couple relationship because relations between women aren’t stable*”.(R., Woman no. 7)

#### 3.4.3. The Bat-Kol Organization

As stated in the organization’s website, “Bat Kol is an Israeli organization for lesbians who are Orthodox Jews. The organization was founded to provide a home for religious women struggling to reconcile their traditional religious way of life and their sexual orientation.” All the women in the sample were members of the organization, which was highly present and significant in their lives. The degree of their involvement in the group differed according to the family structure of each woman, her family and couple status, and the point she had reached in her self-identification as a lesbian. The participants described how their engagement with the organization had changed over time, relating in particular to the importance of Bat-Kol in giving legitimacy to their identity and way of life and imbuing their daily struggle with new meaning.

“*For me the Bat-Kol organization was a lifesaver. I felt like a person going into a dark woods and at the far end there was a cabin with lights on called Bat-Kol. If I knocked on the door they would welcome me in with love, but in the meantime I wanted to walk through the woods alone. At that stage I was an ultra-Orthodox activist. For me, coming out of the closet meant abandoning my public life in favor of my private or sexual life. It didn’t suit me…I defined myself as asexual. I was so much in denial that I volunteered with the Bat-Kol organization and called myself “the straight friend of lesbian women*”.(M., Woman no. 4)

The women meet once a month and receive acknowledgement and support from other women in similar situations. G. (Woman no. 6), who has been in a support group for several years, related to the impact of her activities in Bat-Kol beyond her personal coping.

“*I’m very active in Bat-Kol today. I hope there will be more support for women in the sector so they don’t have to suffer the way I did. I also hope they’ll understand that there has to be special legal counseling for women in the ultra-Orthodox community because when they divorce they go through hell. They’re regarded as rebelling against religion as well as against their husband. If my story can help one woman, I will have done enough.*”

### 3.5. Theme 5. Alternative Family Structures: Resolving the Conflict

The participants in our study sought ways to resolve the conflict between their identities, wishing to continue to live a normative ultra-Orthodox family and community life while remaining true to their sexual identity. They found the solution in a central value of their society—the family. Together with their husbands and female partners, they created unique family structures that enabled them to maintain an overt ultra-Orthodox identity and covert lesbian identity at one and the same time. We identified two such structures each of which had its costs and benefits.

#### 3.5.1. Married with Two Couple Relationships (Women Nos. 2, 3, 4, 6)

This alternative structure is described in the following remarks:

“*I’m in a relationship with a married ultra-Orthodox woman but her husband doesn’t know. For his part, my husband gives me my freedom. He knows, and accepts that that’s who I am. We’re very good friends, but we’re not attracted to each other*”.(N., Woman no. 6)

B. (Woman no. 2) noted what she considered the benefits of this model.

“*It’s convenient to be married. Starting to make custody arrangements is a problem. It’s great for me that he’s at home. He’s very involved in the family, active, takes care of the children, doesn’t care if I bring a girlfriend home. We have an agreement: once every four weeks I’m not at home. I’m with girlfriends…Being married is also good for our image: father, mother, children.*”

On the other hand, H. (Woman no. 3) spoke of the costs of this family structure for herself and her husband.

“*I have a girlfriend, a partner, and we get together. My husband knows but he doesn’t know who she is. It’s a problem. He gets mad at me for suddenly leaving him trapped. I don’t ask what he does. I’d rather hide…I see my husband in the community and suffering very much, and at the same time he lives a life of deep faith and he has to keep it a secret. He has no one to talk to. No one would accept him. We’re both victims.*”

Later in the interview she stated:

“*The risk I would be taking by leaving is very great. First of all, he claims that the rabbinical texts say I’m a rebellious woman, rebelling against both the religion and my husband. I’m not entitled to anything. I’m afraid he’d take the children.*”

#### 3.5.2. Divorced with a Secret Lesbian Relationship (Women Nos. 1, 5, 7)

In the second family structure, the women felt they could no longer remain married. The major benefit of divorcing their husbands was that it made it possible for them to give expression to their lesbian identity. However, their desire to conform to the norms of ultra-Orthodox society led to the high price they pay for their decision. M. (Woman no. 5) described the efforts she and her husband made to avoid a divorce.

“*I didn’t want to come out of the closet and cope with what people would think, and I was also afraid of what would happen to the children. My husband started to see another woman, but he didn’t want to lead a secret life. What broke the camel’s back was when I met my current partner…Both of us got divorced after conducting parallel relationships for a long time. Today my situation is surreal. I live in the same house with my partner and my ex-husband.*”

To the outside world, M. and her ex-husband are still living together for family reasons and are renting a room to the other woman. Inside the home, the situation is different: the man occupies one room, and the women share another.

R. (Woman no. 7) explained that even after divorcing, a woman cannot reveal her lesbian identity.

“*My ex-husband begged me not to get a divorce. He was willing to stay with me, even thought maybe he hadn’t treated me well enough. It didn’t even occur to him that I might really be a lesbian. There’s no such thing in our society. If it was up to him, he would have stayed, even if it cost him. He asks me and our daughters not to talk about it. He says, “Treat it as if it didn’t happen.”*”

A. (Woman no. 1) related that, unlike her, her previous female partner chose not to get a divorce because of her fear of social sanctions. However, she went on to admit that although she did divorce her husband, she shared the same fears.

“*There are women who remain married to their husbands. That’s their choice. I once had a partner who was married and also in a relationship with me. She said she’d never get a divorce. She said it was both from financial fear and fear of what people would say and how it would affect the children’s matchmaking chances. From my point of view, that’s being weak. On the other hand, when matches are made for my children, no one will know about me. Just that I’m divorced.*”

## 4. Discussion

This study drew on social representation theory [5], which contends that the individual’s social representations are constructed in a constant dialogue among the members of the group to which they belong [6] and serve as a guild for action throughout life. The theory distinguishes between social representations in modern societies, which can be diverse and even conflicting, and those in traditional societies, where multiple identities are possible only if they are internally consistent. The study examined members of a conservative traditional society, exploring how lesbian women in the Jewish ultra-Orthodox sector in Israel attempt to maintain their religious identity in order to preserve their sense of belonging to the community while at the same time acknowledging and maintaining their sexual identity.

### 4.1. Social Construction of the Ultra-Orthodox Identity

The interviews we conducted revealed the significance of the social structures in ultra-Orthodox society—the family of origin, the education system (from pre-school to the seminar for girls), and the institution of matchmaking—which serve as agents of socialization for the conventional religious identity. All the women in the study noted that the family and all-girls’ high school (seminar) stressed the society’s highest values: religion and the family [2]. They related how, in the classic process of construction of the identity of the ultra-Orthodox woman, the social rules pave the way for the development of sexual identity in tandem with religious identity. From an early age, a girl is set on the road to marriage with a man “from a good home,” a religious scholar who faithfully observes the commandments. Throughout her life, a girl learns, whether through modeling or through active instruction, how to be the wife of a man for whom “religious study is his trade,” and how to raise pious children. The matchmaking process is a major marker of the ultra-Orthodox identity. The instructions given brides in anticipation of their wedding night provides the young woman with the rules for intimacy with her husband according to religious law that will lead to creating a family. The interviews indicate that at an early stage in their lives, the women all took on themselves the isolation of the ultra-Orthodox community from “other” societies, that is, both secular Israeli society in all its diversity and non-ultra-Orthodox religious society. The sense of belonging derived from embracing the ultra-Orthodox identity comes at the price of submission to rabbinical authority and conforming with the behavioral codes that dictate every aspect of the life of the individual and family [16]. Interestingly, in describing the socialization mechanisms in ultra-Orthodox society, some of the women in our sample reported feeling a lack of a sense of belonging as early as childhood and adolescence, whether because of their ethnicity (as their society regards Ashkenazi origins as superior to Sephardic) or because their parents had returned to religion, also considered lower in status than those born and raised in the community. In the course of the interviews, when these women referred to their lack of a sense of belonging stemming from their lesbian identity, they recalled their similar childhood experience of alienation.

### 4.2. Development of Sexual Identity

The women’s sexual identity was also shaped by the behavioral norms of ultra-Orthodox society [6] and its agents of socialization. The interviews indicate that discourse in both the family and the seminar is two-dimensional, relating to the overt and the covert. All the participants stated that they were taught to hide their sexual development in the name of modesty, a central value in their society [13]. When one of the interviewees found an anatomy book, she was told bluntly by her mother, “Righteous people don’t do that.” The seminar, a potential site of early experimentation with sex and sexuality, provided no guidance on the subject. Thus, relations of a sexual nature between girls were regarded as “close friendships.” Nevertheless, the girls sensed that what they were doing was forbidden, even though they did not know the term “lesbian.” This situation is in line with Foucault, who linked knowledge, power, and discourse, contending that discourse is a system of knowledge that allows some things to be said and disallows others [43]. Discourse imposes its power on the subject by virtue of its ability to determine the truth the subject must acknowledge. Ultra-Orthodox society creates discourse aimed at reinforcing its values. Menstruation, for example, is not associated with sexuality, but with the value of bringing children into the world [32]. The sharp contrast between the absence of discourse on sexuality, sexual desire, and romantic love on the one hand, and sexual relations conducted according to religious law on the other, demonstrates the duality of the discourse of overt and covert. On the overt level, there is a conspiracy of silence surrounding sexuality throughout a girl’s childhood and adolescence until she is ready for the formal guidance given to brides before their wedding. Here, she is provided with concrete instructions meant to prepare her for intimacy with her husband on her wedding night and thereafter, with the emphasis on observance of religious precepts.

### 4.3. Overt and Covert in the Couple Relationship

The participants’ relationships with their husbands are at the heart of their conflict. The ultra-Orthodox woman is brought up to be the wife of a religious scholar who devotes his life to his studies, and the mother of pious children [18]. According to our interviews, in her first sexual encounter with her husband, a woman who has known for some time that she is not attracted to men is required to repress her same-sex preference. Those who are not yet aware of their sexual orientation do not understand why they are not attracted to their husbands. The lack of attraction was reflected openly in the participants’ couple relationships. In one case, the couple did not have sexual relations on the wedding night but only talked, like friends. Another participant described how she recoiled whenever her husband touched her, and a third called out her girlfriend’s name during intercourse, but her husband pretended not to hear. Eventually, she developed vestibulitis (Vestibulitis, otherwise known as “localized provoked vulvodynia”, was first recognized in the late 1980s by gynecologist Edward Friedrich. It is characterized by a stinging or burning-like pain at the vaginal introitus that is provoked by sexual intercourse and the insertion of objects such as a tampon or speculum into the vagina. Vestibulitis usually develops between the ages of 20 and 50 years, often following an infection of the lower genital tract [44] and sexual relations ceased entirely. Women who continued to have intercourse with their husbands employed mechanisms of repression, denial, and detachment in order to silence their true desires, resulting in a sense of loneliness and distress. Thus, for example, one participant convinced herself that the problem lay not in a lack of attraction to her husband but in the fact that she was asexual (Being asexual means lacking sexual attraction to others, or possessing a low interest in sexual activity. Some people consider asexuality to be their sexual orientation, and others describe it as an absence of sexual orientation (https://www.webmd.com/sex/what-is-asexual (accessed on 13 June 2022)))). Another woman turned to the conventional religious practice of prayer (Mansfeld et al., 2016), asking God for the strength to rid her of her attraction to women. All the interviewees stated that they preoccupied themselves with the home and children, thereby distracting their minds from thoughts of their silenced sexual identity. At this point in their lives, the women were conducting family lives that appeared to conform with the norms of ultra-Orthodox society. Secretly, however, they were beginning to recognize the need to create a different family model. The growing sense of a lack of inner peace and discomfort in the spousal relationship ultimately led them to create alternative family structures that enabled them to bridge the gap between their two identities.

### 4.4. Children, Extended Family, and Community

Over the course of their marriage, the women’s lesbian identities became an open secret to themselves and their husbands and remained concealed from their surroundings, including their children, extended family, and community. The issue of how and to whom to disclose their secret arose in all the interviews. In respect to the children, two patterns emerged, both influenced by the social norms regarding relations between women. Thus, some women chose not to tell their children because they attended ultra-Orthodox schools. Knowledge of their mother’s sexual orientation might make them uncomfortable or, even worse, should it become known, they might be expelled from school and consequently sully the family’s good name and impair their chances of making a good match in the future. Other women revealed their secret to their children, saying, for example, that Mom loves Dad and women too. All the participants noted that although there was no religious ban on relations between women, lesbianism was not recognized in their society.

The same duality of disclosure and concealment was seen in respect to the extended family. In this context as well, the women did not reveal their secret, although they all reported sensing that family members knew but collaborated in keeping it hidden. Like the ultra-Orthodox community at large, the family adopted the narrative that women could not be sexually attracted to each other, but only good friends. As one of the interviewees stated, “The family doesn’t regard me as a lesbian, but as a woman who likes women.” Thus, even at this stage in the women’s lives, the extended family continues to function as an agent of socialization of ultra-Orthodox society, believing (whether consciously or not) that they are thereby protecting themselves, their daughter, and her children from ostracization. By accepting the narrative that denies the existence of lesbianism, the family is unable to provide the woman with an open support system. This behavior is consistent with the precedence given in this society to the welfare of the community over the needs of the individual [1].

The women also described how the religious–community system reacted to their lack of attraction to their husband. When they followed the accepted practice in their community and turned to a rabbi or counselor for help [23], they were met with an offensive and violent response and the use of various means of force. Although this issue is beyond the scope of the current research, it is important for future studies to consider the role of these counselors as a further factor that reenforces the norms and values of ultra-Orthodox society [28] and silences the inner voices of the women who seek their assistance. In contrast to the community, Bat-Kol served as a source of support to the interviewees and allowed their voice to be heard. The organization’s mission statement, defining its role as aiding women who wish to be true to both their religious and lesbian identities, constituted the basis for the participants’ decision to create alternative family structures.

### 4.5. Alternative Family Structures

Like the gay men in an American study [39], the women in our sample cannot imagine a life outside the ultra-Orthodox community. Unwilling to reveal their lesbian orientation so as not to hurt their children’s future, they find themselves with two conflicting identities. However, whereas the religious identity is clear for all to see, the sexual identity remains hidden. As a conservative society, the ultra-Orthodox community is not tolerant toward inconsistent identities. The women therefore had to devise ways to reconcile the two. They found the solution in alternative family structures. Two models were described. In the first, the women remained married but had two couple relationships; in the second, they divorced, but continued to conceal their lesbian relationship. Both models can be considered defaults, that is, they do not enable disclosure of the woman’s sexual identity, but only allow it to exist side by side with her religious identity. While this partial solution has its benefits, it also has costs for the woman, her husband, and her children. Figure 1 presents the process the women underwent in creating the alternative family structure.

#### 4.5.1. Model 1: Overt: Married, Covert: Two Couple Relationships

Four participants in our study described this structure. Each of them lives with her husband and also has a lesbian relationship with the husband’s knowledge. The women dubbed this an “open” secret and consensual arrangement (albeit the husband’s consent was given unwillingly, and years after the secret lesbian’s relationship had started, as both sides feared the social consequences of divorce). One of the interviewees had been living in this manner for over eleven years. While this model enables the married couple to maintain the appearance of a normative religious family and therefore not to incur the potential consequences of the woman’s sexual orientation for her children, it takes a heavy psychological toll. The wife is forced to silence her sexual identity for many years, to lead a double life, and to invest considerable physical and mental energy in keeping her secret. The husband pays an even higher price. The women describe him as angry and frustrated, and harboring a sense of being cheated on and being trapped in the marriage. Unlike the woman who is in another relationship in which she is loved and desired, the husband is left feeling lonely and rejected by his wife and cannot turn to anyone in the community for support. As one of the participants put it, “We’re both victims.”

#### 4.5.2. Model 2: Overt: Divorced, Covert: Being in a Lesbian Relationship

Three participants adopted this structure, in which the couple decided to divorce. This is always a difficult decision, but it is even harder in ultra-Orthodox society. Statistics from 2020 show a 4% divorce rate in this sector in Israel, as compared to 16% in the rest of the population [45]. The ultra-Orthodox regard divorce as an infringement of the precept of the integrity of the family unit, and therefore to be avoided at all costs [24]. The decision to get a divorce despite this attitude might be expected to allow the women to openly express their sexual identity. However, this is not the case. Rather, all the women who opted for this model related to the fear of the social sanctions they would suffer if it were to become known, sanctions that would impact their economic status and their children’s education and future matches. One woman continued to live with her ex-husband and her female partner under the same roof under the pretext that she was a tenant. Another woman could not disclose her sexual identity because of her ex-husband’s request that she “treat it as if it didn’t happen.” The third woman explained that although she was divorced, her partner was not, and so their relationship had to remain a secret. Thus, here, too, it is clear that both the husband and wife pay a heavy price for the inability of ultra-Orthodox society to accept inconsistent identities.

## 5. Limitations

The major limitation of the study lies in the fact that all the participants were recruited through Bat-Kol. Their very membership in the organization indicates that they have internalized their lesbian identity and processed it sufficiently to enable them to discuss it openly, if not with their surroundings, at least with other women in a similar situation. It is possible that their activity in Bat-Kol was one of the factors that led them to feel secure and strong enough to create alternative family structures to bridge between their two conflicting identities. However, the nature of recruitment did not allow us to learn about the experience of lesbian ultra-Orthodox women who do not take advantage of the Bat-Kol support groups, or what, if any, solutions they have found to their conflict. Nor do we know whether the results also characterize lesbian ultra-Orthodox women outside Israel. Future studies conducted in other countries might shed light on the family structures created by ultra-Orthodox families elsewhere in the world in which one or more family member is coping with conflicting identities.

In addition, as noted above, the study was conducted from the interpretative perspective of Israeli researchers who are social workers and secular heterosexual mothers. Despite our best efforts to understand the phenomenon through the eyes of the participants themselves, it is not inconceivable that “noises” from our own culture of which we were unaware influenced our interpretation of the findings. It is our hope that our reflective abilities as researchers and professional social workers enabled us to convey the women’s experience as faithfully as possible.

## 6. Conclusions

Despite these limitations, the study makes an important contribution to both the theoretical and the practical literature. On the theoretical level, it extends the social representations theory to inconsistent identities in ultra-Orthodox society, specifically the inconsistency between religious identity and lesbian sexual identity. Furthermore, the study offers a schematic model of the way in which these conflicting identities are bridged by means of alternative family structures.

The study also expands the discourse on silencing in ultra-Orthodox society. This issue is generally discussed in reference to sexual abuse or domestic violence [1]. The current research is the first to add the context of lesbian identity.

On the practical level, the study helps give a voice to lesbian women in the ultra-Orthodox community and the complexity of their lived experience. The interviews reveal the processes with which they must cope, the resources available to them (Bat-Kol, alternative family structures), and the factors that are liable to impair their mental welfare (inappropriate counselling, imposition of a traditional lifestyle). Through the words of the women, the distress of their spouses and children also becomes apparent. This information can serve as the basis for professional interventions aimed at assisting these families to deal with the difficulties inherent in ultra-Orthodox society, in the form of individual or family counselling, as well as legal aid when necessary. The findings may also promote counselling for young women before marriage, at the stage when they are attempting to define their sexual identity.

## Figures and Tables

**Figure 1 ijerph-19-07575-f001:**
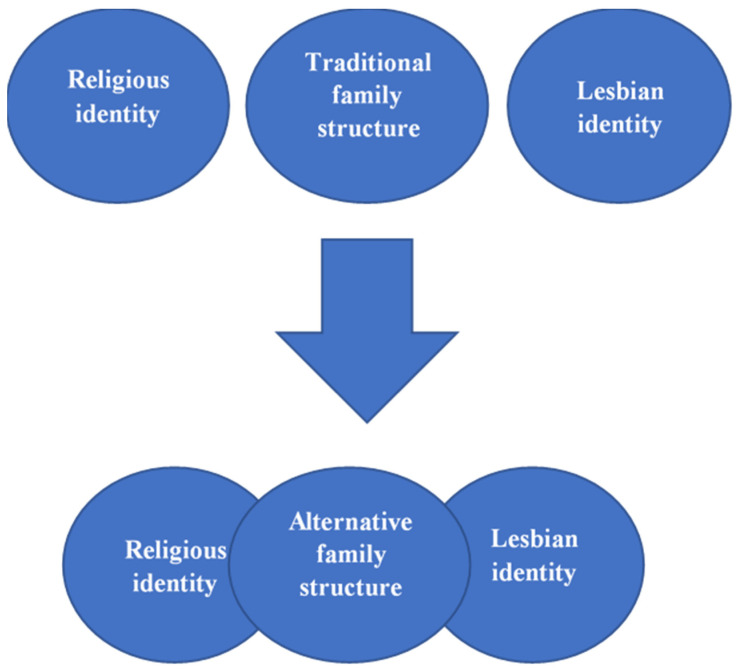
Alternative family structures as a bridge between identities.

**Table 1 ijerph-19-07575-t001:** Characteristics of the participants.

	Woman No. 1	Woman No. 2	Woman No. 3	Woman No. 4	Woman No. 5	Woman No.6	Woman No. 7
Family structure	Divorced, secret lesbian relationship	Divorced, secret lesbian relationship	Married, two couple relationships	Married, two couple relationships	Married, two couple relationships	Married, two couple relationships	Divorced, secret lesbian relationship
Number of children	3	7	5	4	5	5	5
Woman’s age	33	59	41	41	42	36	48
Men’s age	33	60	42	41	42	36	48
Residence	Jerusalem	Jerusalem	Bnei-Brak	Bnei-Brak	Jerusalem	Jerusalem	Bat-Yam

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
