# Peer review of "Ultra-Orthodox Lesbian Women in Israel: Alternative Family Structures as a Bridge between Religious and Sexual Identities"

_ijerph, 2022, doi:10.3390/ijerph19137575_

Round 1

Reviewer 1 Report

Ultra-Orthodox Lesbian Women in Israel: Alternative Family Structures as a Bridge between Religious and Sexual Identities

 This study examined the family structures that lesbian ultra-Orthodox women in Israel have adopted to accommodate their conflicting identities. The study employed a qualitative design, conducting in-depth interviews with seven ultra-Orthodox lesbian women. Findings revealed that in all cases, lesbian identity was kept hidden, and expose the unique family structures these ultra-Orthodox lesbian women created that allowed them to maintain their religious way of life on the surface, while remaining committed to their sexual identity in secret.

 The experience of lesbian ultra-Orthodox women navigating between their religious and sexual identities, which was the focus of the current study, has never previously been systematically investigated. Therefore, this study has important contribution to an understudied topic.

 The Introduction and literature review are clear and coherent.

The methodology of the study is adequate to the purpose and very well described.

Regarding recruitment: As this is a very unique sample, where there any difficulties in recruitments? Did the authors approach more participants? or did they approach seven women which all agreed to participate? The authors may consider mentioning this information on page 6.

 The results are very clear, vast, and read easily. They allow an important and authentic glimpse into the experiences of the participants.

 The discussion is well-written and included very clear models summarizing the results. Limitations and implications are clear and concise.

 Thank you for the opportunity to review this very interesting and important paper.

Author Response

Comment: The Introduction and literature review are clear and coherent.

Response:  Thank you for your constructive feedback.

Comment: The methodology of the study is adequate to the purpose and very well described.

Response: Thank you

Comment: Regarding recruitment: As this is a very unique sample, where there any difficulties in recruitments? Did the authors approach more participants? or did they approach seven women which all agreed to participate? The authors may consider mentioning this information on page 6.

Response: Indeed, recruitment was very challenging. As we now describe in the text, we started with a prior acquaintance and used snowball sampling from there. Two of the women initially wanted us to use pseudonyms, but further along, felt fine with revealing their real names.

Comment: The results are very clear, vast, and read easily. They allow an important and authentic glimpse into the experiences of the participants.

Response: Thank you for this feedback.

Comment: The discussion is well-written and included very clear models summarizing the results.

Limitations and implications are clear and concise.

Response: Thank you very much.

Reviewer 2 Report

Good job, the paper is insightful and informative. I just have some minor comments that need to be addressed.

1. Bat-Kol served as a support group for lesbians, both closseted and out. I am curious if the nature of the organization is know and if yes, how does society view the organization and its members. I recommend that a brief description about Bat-Kol be added either in the introduction or method section.

2. The names (naming) of the sub-themes are not consistent. From the general discussion/introduction of the 5 themes and its sub-themes at the beginning of the Results section to the detailed discussion thereafter. I recommend that you be consistent so as not to confuse the reader.

3. How is theme 2 (Family & Educational System) different from the sub theme (nuclear family & educational system) in Theme 1 (Macro System)?

4.  Describe the terms, vestibulitis and asexual since not everyone is familiar with these terms.

Author Response

Comment: Good job, the paper is insightful and informative. I just have some minor comments that need to be addressed.

Response: Thank you for your feedback.

Comment: Bat-Kol served as a support group for lesbians, both closs Feted and out. I am curious if the nature of the organization is know and if yes, how does society view the organization and its members. I recommend that a brief description about Bat-Kol be added either in the introduction or method section.

Response: Thank you for this important comment. We wrote about Bat-Kol on Page 27 (sub-theme 4.3), in the section about our findings. We realized, however, that we need to address this issue much earlier in the manuscript and now include information about the organization in the Method section, on pages 10-11.

Comment:  The names (naming) of the sub-themes are not consistent. From the general discussion/introduction of the 5 themes and its sub-themes at the beginning of the Results section to the detailed discussion thereafter. I recommend that you be consistent so as not to confuse the reader.

 Response: Thank you, you are right, and the themes were not consistent. We fixed it now and added numbers next to each of the sub-themes, to make it easier to follow.

Comment:  How is theme 2 (Family & Educational System) different from the sub theme (nuclear family & educational system) in Theme 1 (Macro System)?

Response:  Thank you for pointing this out. Following your previous comment, we reorganized all the themes and sub-themes and made sure to correct their names where necessary. We believe that the confusion was due to our misuse of themes’ headlines. We hope the distinction among themes is now clearer. Theme 1: The Macro System: Social Construction of the Religious Identity; and Theme 2: The Family and Educational System: Ignorance, Secrecy, and Repression of Women’s Sexuality.

Comment:  Describe the terms, vestibulitis and asexual since not everyone is familiar with these terms.

Response: We have now added footnotes to explain these terms.

Reviewer 3 Report

Thank you for an interesting and engaging article on an important and  understudied topic. The writng was clear and concise, the methodology appropriate and well-applied, and the results described with clarity and insight. I have no criticisms to make.

Author Response

Comment:  Thank you for an interesting and engaging article on an important and understudied topic. The writing was clear and concise, the methodology appropriate and well-applied, and the results described with clarity and insight. I have no criticisms to make.

Response: Thank you for your support.